# On Representing Convex Quadratically Constrained Quadratic Programs via Graph Neural Networks

## Abstract

Convex quadratically constrained quadratic programs (QCQPs) involve finding a solution within a convex feasible region defined by quadratic constraints while minimizing a convex quadratic objective function. These problems arise in various industrial applications, including power systems and signal processing. Traditional methods for solving convex QCQPs primarily rely on matrix factorization, which quickly becomes computationally prohibitive as the problem size increases. Recently, graph neural networks (GNNs) have gained attention for their potential in representing and solving various optimization problems such as linear programs and linearly constrained quadratic programs. In this work, we are the first to investigate the representation power of GNNs in the context of QCQP tasks. Specifically, we propose a new tripartite graph representation for general convex QCQPs and properly associate it with message-passing GNNs. We demonstrate that there exist GNNs capable of reliably representing key properties of convex QCQPs, including *feasibility*, *optimal value*, and *optimal solution*. Our result deepens the understanding of the connection between QCQPs and GNNs, paving the way for future machine learning approaches to efficiently solve QCQPs.

## 1 Introduction

*Quadratic programs* (QPs) are a pivotal class of optimization problems where the objective function is quadratic, and the constraints are typically linear or quadratic. Based on the nature of constraints, QPs can be further classified as *linearly constrained quadratic programs* (LCQPs) and *quadratically constrained quadratic programs* (QCQPs). When the objective and constraint matrices are positive semi-definite, the problem becomes a convex QCQP, making it both theoretically interesting and practically important. Convex QCQPs arise in various critical applications such as robust optimization in uncertain environments (Ben-Tal & Nemirovski, 2001; Boyd & Vandenberghe, 2004), power flow (Bienstock et al., 2020), and signal processing (Luo et al., 2010), while ensuring optimality and computational efficiency is paramount.

Solving QPs, especially those with quadratic constraints, presents significant challenges. Traditional methods often involve computationally intensive procedures that would struggle with scalability and real-time processing requirements. For example, the *interior-point method* (Nocedal & Wright, 1999) for a general $n$-variable QP involves solving a sequence of linear systems of equations, necessitating matrix decomposition with a runtime complexity of $\mathbb{R}^s(n^3)$. This leads to substantial computational burden in the large-scale case. Similarly, active-set algorithms (Gill et al., 2019), which work by iteratively adjusting the set of active constraints, can also become computationally demanding as the number of constraints and variables increase.

In recent years, advances in *machine learning* (ML) have opened new avenues for enhancing the solving process of QPs. There are mainly two categories of ML-aided QP methods. The first category

aims to learn adaptive configurations of a specific QP algorithm or solver to accelerate the solving process (Bonami et al., 2018; Ichnowski et al., 2021; Jung et al., 2022), while the second focuses on predicting an initial solution of QPs, which is either directly taken as a final solution or further refined by subsequent algorithms or QP solvers (Bertsimas & Stellato, 2022; Gao et al., 2021; Sambharya et al., 2023; Tan et al., 2024; Wang et al., 2020). Most of these methods utilize *graph neural networks* (GNNs) to leverage the structural properties of graph-structured data, making them particularly well-suited for representing the relationships and dependencies inherent in QPs. By encoding QP instances into graphs, GNNs can capture intricate features and provide adaptive guidance or approximate solutions efficiently.

In addition to these empirical studies, theoretical research on the expressive power of GNNs (Zhang et al., 2023; Li & Leskovec, 2022) and their relation to optimization problems has further strengthened the understanding of their capabilities. For instance, Chen et al. (2022a) and Chen et al. (2022b) established theoretical foundations for applying GNNs to solving *linear programs* (LPs) and *mixed-integer linear programs*, respectively. Further, such foundations are extended to LCQPs and their discrete variant, mixed-integer LCQPs in Chen et al. (2024).

Previous studies have empirically and theoretically demonstrated the utility of GNNs in speeding up existing QP solvers and directly approximating solutions for various QP instances. However, there is a noticeable lack of research on their use in QCQPs, particularly in how they handle quadratic constraints. *Existing graph representations used for LCQPs (Chen et al., 2024) are inadequate for QCQPs* as they can not capture the complex interactions introduced by quadratic constraints. Moreover, the question of *whether GNNs can accurately predict key properties of QCQPs, such as feasibility, optimal objective value, and optimal solution*, remains open.

This paper aims to address the aforementioned gap by exploring both theoretical foundations and practical implementation of using GNNs for solving convex QCQPs. Specifically, we propose a tripartite graph representation for general convex QCQPs, and establish theoretical foundations of applying GNNs to optimize QCQPs. The distinct contributions of this paper can be summarized as follows.

- **Graph Representation**. We propose a novel tripartite graph representation for general QC-QPs, which divides a QCQP into three types of nodes: linear-term, quadratic-term, and constraint nodes, with edges added between heterogeneous nodes to indicate problem parameters. This representation effectively addresses the limitations of existing graph representations for LCQPs, i.e., those graphs are unable to capture the interactions imposed by quadratic constraints.

- **Theoretical Foundation**. We conduct analysis on the *separation power* as well as *approximation power* of *message-passing GNNs* (MP-GNNs). We showed that MP-GNNs are capable of capturing some key properties of convex QCQPs.

- **Empirical Evidence**. We conduct initial numerical tests of the tripartite message-passing GNNs on small QCQP instances. The results showed that MP-GNNs can be trained to approximate the key properties well.

NOTATIONS

Throughout this paper, scalars or vectors are denoted by lowercase letters (e.g., $a$), and matrices are denoted by uppercase letters (e.g., $A$). For a vector $a$, we denote its $i$-th entry by $a_i$. For a matrix $A$, the entry in the $i$-th row and the $j$-th column is denoted by $a_{i,j}$. We use $\mathbf{0}$ and $\mathbf{1}$ to denote vectors or matrices with all-zero and all-one entries, respectively. For any positive integers $m, n$ with $m < n$, we define $[m, n] \coloneqq \{m, m+1, \cdots, n\}$ to be the set of all integers ranging from $m$ to $n$. For brevity, we define $[n] \coloneqq [1 : n] = \{1, 2, \cdots, n\}$.

## 2 GRAPH REPRESENTATION OF QCQPS

### 2.1 QUADRATICALLY CONSTRAINED QUADRATIC PROGRAMS

In this work, we study QCQPs defined in the following form:

$$
\begin{aligned}
\min_{x \in \mathbb{R}^n} \quad & \frac{1}{2}x^\top Q x + p^\top x \\
\text{s.t.} \quad & \frac{1}{2}x^\top Q^i x + (p^i)^\top x + b^i \le 0 \qquad \forall\, i \in [m] \\
& x^{\mathrm{L}} \le x \le x^{\mathrm{U}}
\end{aligned}
\tag{2.1}
$$

where $Q, Q^i \in \mathbb{R}^{n \times n}$, $p, p^i \in \mathbb{R}^n$, $b^i \in \mathbb{R}$, $x^{\mathrm{L}} \in (\mathbb{R} \cup \{-\infty\})^n$, and $x^{\mathrm{U}} \in (\mathbb{R} \cup \{+\infty\})^n$. The problem has $n$ optimization variables and $m$ constraints. We refer to the tuple $(m, n)$ as the *problem size* of QCQP. Both the objective function and the constraints are associated with quadratic functions. Without loss of generality, we assume $Q$ and $Q^i$'s are all symmetric matrices. The QCQP problem is *convex* if $Q$ and $Q^i$'s are all positive semidefinite.

We denote the *feasible set* of Problem 2.1 by

$$
\mathcal{X} := \left\{ x \in \mathbb{R}^n : \quad \frac{1}{2}x^\top Q^i x + (p^i)^\top x + b^i \le 0, \ \ \forall i \in [m], \ \ x^{\mathrm{L}} \le x \le x^{\mathrm{U}} \right\}.
\tag{2.2}
$$

If $\mathcal{X} \ne \emptyset$, the QCQP is said to be *feasible*; otherwise, it is said to be *infeasible*. A feasible QCQP is said to be *bounded* if the objective is bounded from below on $\mathcal{X}$, i,e., there exists $z \in \mathbb{R}$ such that $\frac{1}{2}x^\top Q x + p^\top x \ge z$ for every $x \in \mathcal{X}$; otherwise, it is said to be *unbounded*. For a feasible and bounded QCQP, $x^* \in \mathcal{X}$ is said to be an optimal solution if

$$
\frac{1}{2}x^{*\top} Q x^* + p^\top x^* \le \frac{1}{2}x^\top Q x + p^\top x
\tag{2.3}
$$

for every $x \in \mathcal{X}$. We remark that a QCQP always admits an optimal solution if it is feasible and bounded, but such an optimal solution might not be unique.

### 2.2 TRIPARTITE REPRESENTATION OF QCQPS

The first theoretical result demonstrating the representation power of GNNs in solving optimization problems was provided by Yin et al. Chen et al. (2022a). In this work, the information of an LP problem is encoded into a bipartite graph, where variables and constraints are modeled as nodes, and their association is represented as edges. They showed that GNNs based on this graph representation can universally approximate the optimal solution of LPs, as well as properties of feasibility and boundedness. This bipartite graph modeling was later extended to analyze the representation power of GNNs for LCQPs (Chen et al., 2024).

Despite these advances, it remains challenging to develop graph representation to encode all information of general QCQPs while maintaining simplicity for GNN processing. Due to the presence of quadratic terms, a QCQP generally involves $O(n^2 \times m)$ coefficients. Consequently, a graph encoding all QCQP information inherently exhibits a complexity of the same order, $O(n^2 \times m)$. There are two natural extensions of the traditional bipartite representation of LP/LCQP to QCQP.

- **Hyperedge Representation**. This approach adds hyperedges to the traditional bipartite graph to represent quadratic coefficients, turning the graph into a hypergraph. However, to the best of our knowledge, current GNN architectures struggle to handle hyperedges efficiently.

- **Vector Feature Representation**. In this method, all coefficients are encoded as features associated with the $n$ variable nodes and the $m$ constraint nodes, resulting in a graph with vector features of varying sizes, depending on the problem. However, existing GNNs are generally incapable of processing features of varying dimensions.

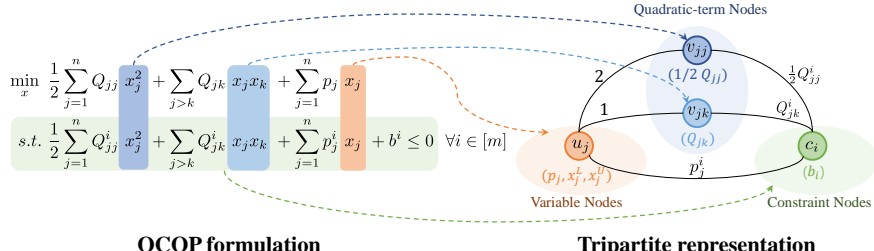

Figure 1: A tripartite representation of QCQPs. It consists of three types of nodes: variable nodes, quadratic-term nodes, and constraint nodes. All nodes and the edges connecting them are associated with coefficients from the formulation as features.

To fill this gap, we introduce an undirected *tripartite graph* representation $G_{\text{QCQP}} := (V, E)$ that encodes all elements of a QCQP (2.1). Compared to the traditional bipartite graph modeling for LPs and LCQPs, our tripartite graph representation introduces an additional class of nodes to model the quadratic terms of variables. This modification allows us to represent QCQPs without any loss of information. In this paper, we will show that the tripartite representation enables GNNs to universally approximate solutions for convex QCQPs.

Formally, a tripartite graph modeling a QCQP consists of three types of nodes, representing variables, constraints, and quadratic terms respectively. Specifically, we define $V_1 := \{u_1, u_2, \ldots, u_n\}$ as the set of nodes where each $u_i$ corresponds to the variable $x_i$. Each node $u_i$ is associated with a feature tuple $(p_i, x_i^{\text{L}}, x_i^{\text{U}})$. Next, we define $V_2 := \{v_{j,k} : (j, k) \in \mathcal{L}\}$ as the set of nodes representing the quadratic terms, where $\mathcal{L} := \{(j, k) \in [n] \times [n] : j \le k, |q_{j,k}| + \sum_{i \in [m]} |q_{j,k}^i| > 0\}$. We remark that if $(j, k) \in L$, the coefficient of the quadratic term $x_j x_k$ is non-zero in the objective function or at least one of the constraints. For each node in $V_2$, if $j > k$, $v_{i,j}$ is associated with a feature $2q_{j,k}$; if $j = k$, $v_{j,j}$ is associated with a feature $q_{j,j}$. Further, we define $V_3 := \{c_1, c_2, \ldots, c_m\}$ as the set of nodes where each $c_i$ represents the $i$-th constraint, with each node $c_i$ associated with a feature $b_i$. Therefore, the set of all nodes in the QCQP graph is given by $V := V_1 \cup V_2 \cup V_3$.[1]

The QCQP graph also includes three types of edges. Let $E_{12} := \{(u_{j'}, v_{j,k}) \in V_1 \times V_2 : j' = j \vee j' = k\}$ be the set of edges connecting nodes from $V_1$ to those in $V_2$. The weight of an edge $(u_{j'}, v_{j,k})$ is 1 if $j > k$ and 2 otherwise. Let $E_{13} := \{(u_j, c_i) \in V_1 \times V_3 : p_j^i \ne 0\}$ be the set of edges connecting nodes from $V_1$ to those in $V_3$. The weight of an edge $(u_j, c_i)$ is $p_j^i$. Let $E_{23} := \{(v_{j,k}, c_i) \in V_2 \times V_3 : q_{j,k}^i \ne 0\}$ be the set of edges connecting nodes from $V_2$ to those in $V_3$. The weight of an edge $(v_{j,k}, c_i)$ is $2q_{j,k}^i$ if $j > k$ and $q_{j,j}^i$ otherwise. Thus, the set of all edges is given by $E := E_{12} \cup E_{13} \cup E_{23}$. Throughout this paper, we denote the weight of the edge between $u \in V_1$ and $v \in V_2$ by $w_{u,v} = w_{v,u}$, and similarly $w_{u,c} = w_{c,u}$ for edges between $V_1, V_3$, $w_{v,c} = w_{c,v}$ for edges between $V_2, V_3$.

We illustrate this representation in Figure 1. We remark that there is a one-to-one mapping between a QCQP and its tripartite graph representation $G_{\text{QCQP}}$.

**Definition 1** (Spaces of Convex QCQP-graphs). *We denoted by $\mathcal{G}_{\text{QCQP}}^{m,n}$ the set of tripartite graph representations for all **convex** QCQPs with $n$ variables and $m$ constraints.* [2]

---

[1] We always denote a variable node by $u$, a quadratic node by $v$, and a constraint node $c$. We always index the constraint nodes by $i$, and the variable/quadratic nodes by $j, k$, unless otherwise specified.

[2] For any QCQP graph in $\mathcal{G}_{\text{QCQP}}^{m,n}$, the associated convex QCQP can be characterized by its coefficient tuple $(Q, \{Q^i\}_{i=1}^m, p, \{p^i\}_{i=1}^m, \{b^i\}_{i=1}^m, x^{\text{L}}, x^{\text{U}}) \in (\mathbb{S}_+^n)^{m+1} \times \mathbb{R}^{n \times (m+1)} \times \mathbb{R}^m \times (\mathbb{R} \cup \{-\infty\})^n \times (\mathbb{R} \cup \{+\infty\})^n$, where $\mathbb{S}_+^n$ denotes the space of positive semidefinite matrices of dimension $n$. We define a topology on $\mathcal{G}_{\text{QCQP}}$: for $Q, Q^i$ and $p, p^i$ we use the topology induced by the norm of the linear mappings defined by the matrices

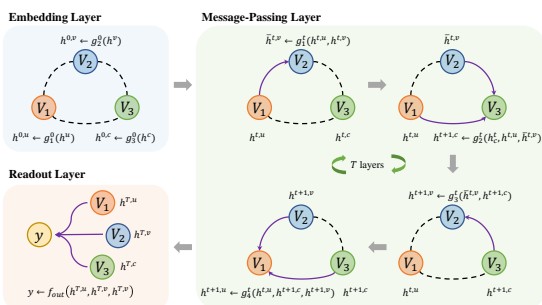

Figure 2: An overview of the GNN architecture.

# 3 THEORETICAL RESULTS

## 3.1 TRIPARTITE MESSAGE-PASSING GNNS

To study the capability of GNNs in representing QCQPs, we tailor the general message-passing GNNs for the tripartite nature of the introduced QCQP graph representation. An overview is depicted in Figure 2. Specifically, we consider the family of tripartite message-passing GNNs consisting of an embedding layer, $T$ message-passing layers (each comprised of four sub-layers), and a readout layer, detailed as follows:

- **Embedding Layer.** For all nodes, the input features $h^{0,u}, h^{0,v}, h^{0,c}$ are initialized by embedding the node features into a hidden space $\mathbb{R}^{h_0}$, where $h_0$ is the space dimension. Specifically,

$$h^{0,u} \leftarrow g_1^0(h^u), \forall u \in V_1 \quad h^{0,v} \leftarrow g_2^0(h^v), \forall v \in V_2, \quad h^{0,c} \leftarrow g_3^0(h^c), \forall c \in V_3$$

  where $g_l^0$'s are learnable embedding functions, $l = 1, 2, 3$, and $h^u, h^v, h^c$ are the node features carried by $u \in V_1, v \in V_2, c \in V_3$.

- **Message-Passing Layer**. Each message-passing layer consists of four sub-layers for updating the features of nodes with learnable functions $f_l^t, g_l^t$. Specifically, each sub-layer updates node features in one of $V_1, V_2, V_3$ by gathering information from certain neighboring nodes.

  - **First sub-layer updating quadratic nodes** ($V_1 \rightarrow V_2$)

  $$\bar{h}^{t,v} \leftarrow g_1^t \left( h^{t,v}, \sum_{u \in V_1} w_{u,v} F_1^t(h^{t,u}) \right), \forall v \in V_2$$

  - **Sub-layer updating constraint nodes** ($V_1 + V_2 \rightarrow V_3$):

  $$h^{t+1,c} \leftarrow g_2^t \left( h^{t,c}, \sum_{u \in V_1} w_{u,c} f_2^t(h^{t,u}), \sum_{v \in V_2} w_{v,c} f_3^t(\bar{h}^{t,v}) \right), \forall c \in V_3$$

  - **Second sub-layer updating quadratic nodes** ($V_3 \rightarrow V_2$):

  $$h^{t+1,v} \leftarrow g_3^t \left( \bar{h}^{t,v}, \sum_{c \in V_3} w_{c,v} f_5^t(h^{t+1,c}) \right), \forall v \in V_2$$

  - **Sub-layer updating variable nodes** ($V_3 + V_2 \rightarrow V_1$):

  $$h^{t+1,u} \leftarrow g_4^t \left( h^{t,u}, \sum_{c \in V_3} w_{c,u} f_5^t(h^{t+1,c}), \sum_{v \in V_2} w_{v,u} f_6^t(h^{t+1,v}) \right), \forall u \in V_1$$

and vectors, and for $x^{\mathrm{L}}, x^{\mathrm{U}}, b^i$ we use euclidean topology on $\mathbb{R}$ and discrete topology on the infinite values. In numerical experiments, we represent the infinite values by introducing an extra infinity indicator.

- **Readout layer** The readout layer applies a learnable function $f_{\text{out}}$ to map the features $h^{T,v}$, $v \in V = V_1 \cup V_2 \cup V_3$ output by the $T$-th (i.e. last) message-passing layer, to a readout $y$ in a desired output space $\mathbb{R}^s$, where $s$ is the dimension of the output space. In this paper, we consider the following two types of output space:

  - Graph-level scalar output ($s = 1$). In this case, we set

$$y = f_{\text{out}} \left( \sum_{u \in V_1} h^{T,u}, \sum_{v \in V_2} h^{T,v}, \sum_{c \in V_3} h^{T,c} \right)$$

  - Node-level vector output with $s = n$. In this case, we only consider the output associated with the variable nodes in $V_1$, given by

$$y_j = f_{\text{out}} \left( h^{T,u_j}, \sum_{k \in [n] \setminus \{j\}} h^{T,u_k}, \sum_{v \in V_2} h^{T,v}, \sum_{c \in V_3} h^{T,c} \right), \quad j \in [n]$$

**Definition 2** (Spaces of GNNs). *Let $\mathcal{F}_{\text{QCQP}}(\mathbb{R}^s)$ denote the collection of all tripartite message-passing GNNs, parameterized by continuous embedding functions $g_{l_1}^0, l_1 = 1, 2, 3$, continuous hidden functions in the message passing layers $g_{l_2}^t, l_2 = 1, 2, 3, 4$, $h_{l_3}^t, l_3 = 1, 2, 3, 4, 5, 6$, and the continuous readout function $f_{\text{out}}$. Specifically, for a given problem size $(m, n)$ of QCQP, there exists a subset of GNNs in $\mathcal{F}_{\text{QCQP}}(\mathbb{R}^s)$ that maps the input space $\mathcal{G}_{\text{QCQP}}^{m,n}$ to the output space $\mathbb{R}^s$. This subset of GNNs are denoted by $\mathcal{F}_{\text{QCQP}}^{m,n}(\mathbb{R}^s)$.*

We define the following target functions, characterizing some key properties on learning an end-to-end network to predict the optimal solutions of convex QCQPs:

**Definition 3** (Target mappings). *Let $G_{\text{QCQP}}$ be a tripartite graph representation of a QCQP problem. We define the following target mappings.*

- *Feasibility mapping: We define $\Phi_{\text{feas}}(G_{\text{QCQP}}) = 1$ if the QCQP problem is feasible and $\Phi_{\text{feas}}(G_{\text{QCQP}}) = 0$ otherwise.*

- *Boundedness mapping: for a feasible QCQP problem, we define $\Phi_{\text{bound}}(G_{\text{QCQP}}) = 1$ if the QCQP problem is bounded and $\Phi_{\text{bound}}(G_{\text{QCQP}}) = 0$ otherwise.*

- *Optimal value mapping: for a feasible and bounded QCQP problem, we set $\Phi_{\text{opt}}(G_{\text{QCQP}})$ to be its optimal objective value.*

- *Optimal solution mapping: for a feasible, bounded QCQP problem, there must exist at least an optimal solution, but the optimal solution might not be unique. However, if the QCQP is convex, there exists a unique optimal solution $x^*$ with the smallest $\ell_2$-norm among all optimal solutions. Therefore, for **convex** QCQP we define the optimal solution mapping to be $\Phi_{\text{sol}}(G_{\text{QCQP}}) = x^*$. Since the optimal solution with the smallest $\ell_2$-norm may not be unique for non-convex QCQP, we do not define its optimal solution mapping.*[3]

### 3.2 UNIVERSAL APPROXIMATION FOR CONVEX QCQPs

Now, we demonstrate that for convex QCQPs, any target function in Definition 3 can be universally approximated by message-passing GNNs. Formally, we have the following theorem.

**Theorem 1.** *For any probability measure $\mathbb{P}$ on the space of convex QCQPs $\mathcal{G}_{\text{QCQP}}^{m,n}$ and any $\delta, \varepsilon > 0$, there exists $F \in \mathcal{F}_{\text{QCQP}}^{m,n}(\mathbb{R}^s)$ such that for any target mapping $\Phi : \mathcal{G}_{\text{QCQP}}^{m,n} \to \mathbb{R}^s$ defined in Definition 3, we have*

$$\mathbb{P}\{||F(G_{\text{QCQP}}) - \Phi(G_{\text{QCQP}})|| > \delta\} < \varepsilon. \tag{3.1}$$

---

[3]In fact, Section 3.3 shows that there exists a pair of non-convex QCQPs that cannot be distinguished by any GNNs. Thus, even if an optimal solution mapping for non-convex QCQPs is defined, GNNs cannot universally approximate it.

$$\min \quad x_1 x_2 + x_2 x_3 + x_3 x_1 + x_4 x_5 + x_5 x_6 + x_6 x_4$$
$$\text{s.t.} \quad \sum_i x_i^2 \leq 1$$

$$\min \quad x_1 x_2 + x_2 x_3 + x_3 x_4 + x_4 x_5 + x_5 x_6 + x_6 x_1$$
$$\text{s.t.} \quad \sum_i x_i^2 \leq 1$$

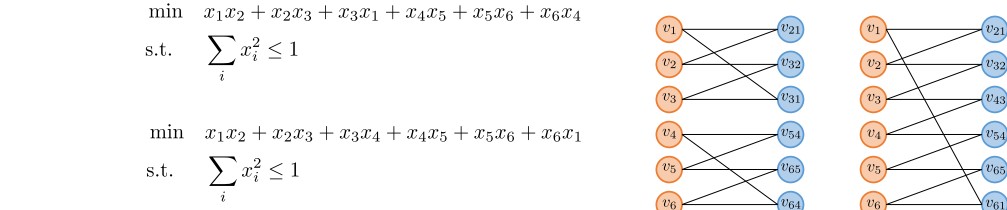

Figure 3: Left: two QCQP instances for proving Prop. 1. Right: Parts of the corresponding tripartite graph representations to show the difference.

Theorem 1 highlights that sufficiently expressive GNNs can predict the feasibility, boundedness, optimal value, and optimal solution for convex QCQP problems with an arbitrarily small error. The proof of Theorem 1 is provided in Appendix A.

### 3.3 MESSAGE-PASSING GNNs CAN NOT REPRESENT NON-CONVEX QCQPs

In contrast to convex QCQPs, message-passing GNNs based on tripartite graph representation do not possess universal representation power for non-convex QCQPs. Formally, we have the following propositions.

**Proposition 1.** *There exists non-convex QCQP instances $\mathcal{I}, \bar{\mathcal{I}}$ encoded by tripartite graph representation $G, \bar{G}$ respectively, such that $\Phi(G)_{feas} \neq \Phi_{feas}(\bar{G})$, but any GNN $F \in \mathcal{F}_{\text{QCQP}}(\mathbb{R})$ gives $F(G) = F(\bar{G})$.*

**Proposition 2.** *There exists non-convex QCQP instances $\mathcal{I}, \bar{\mathcal{I}}$ encoded by tripartite graph representation $G, \bar{G}$ respectively, such that*

1. $\Phi(G)_{opt} \neq \Phi_{opt}(\bar{G})$

2. *the optimal solution sets of $\mathcal{I}$ and $\bar{\mathcal{I}}$ do not intersect*

3. *any GNN $F \in \mathcal{F}_{\text{QCQP}}(\mathbb{R})$ gives $F(G) = F(\bar{G})$.*

Proposition 1 implies that GNNs cannot universally predict the feasibility of non-convex QCQPs. Proposition 2 implies that GNNs can neither universally predict the optimal value nor the optimal solution of non-convex QCQPs. We prove both propositions by constructing counter-examples. Below we present the counter-example for Proposition 2. We defer the formal proof of both propositions to Appendix C.

Consider the following pair of non-convex QCQPs:

$$\min \quad x_1 x_2 + x_2 x_3 + x_3 x_1 + x_4 x_5 + x_5 x_6 + x_6 x_4$$
$$\text{s.t.} \quad \sum_i x_i^2 \leq 1 \tag{3.2}$$

$$\min \quad x_1 x_2 + x_2 x_3 + x_3 x_4 + x_4 x_5 + x_5 x_6 + x_6 x_1$$
$$\text{s.t.} \quad \sum_i x_i^2 \leq 1 \tag{3.3}$$

For the former, the optimal objective value is $\Phi_{\text{obj}} = -\frac{1}{2}$, and all optimal solutions are given by

$$\{x : x_1 + x_2 + x_3 = 0, x_4 + x_5 + x_6 = 0, \sum_i x_i^2 = 1\}.$$

For the latter, the optimal objective value $\Phi_{\mathrm{obj}} = -1$, and all optimal solutions are given by

$$\{x : x_1 = x_3 = x_5 = -x_2 = -x_4 = -x_6 = \pm\frac{\sqrt{6}}{6}\}.$$

We see that the optimal values of Problem 3.2 and Problem 3.3 are different, and their optimal solution sets do not intersect. The tripartite graph representations of the two instances are illustrated in Figure 3. We will further demonstrate in appendix C that any GNN on the two tripartite graphs gives the same output. Thus, Problem 3.2 and Problem 3.3 serve as a valid counter-example for proving Proposition 2.

## 4  COMPUTATIONAL EXPERIMENTS

In this section, we present empirical experiments to validate the proposed theoretical results. The corresponding source code is available at https://anonymous.4open.science/r/l2qp-6B56/.

**Learning tasks:**  In Theorem 1, we established that there exists a fucntion $F \in \mathcal{F}_{\mathrm{QCQP}}^{m,n}(\mathbb{R}^s)$ capable of approximating the target mapping $\Phi$ with an arbitrarily small error. To empirically confirm this claim, we design three supervised learning tasks to find such functions $F_{\mathrm{feas}}$, $F_{\mathrm{obj}}$ and $F_{\mathrm{sol}}$, which are responsible for predicting feasibility, objective values, and optimal solutions, respectively. For each task, a dataset $\{(G_i, y_i)\}_{i=1}^{N}$ is provided, where $G_i$ represents a QCQP instance and $y_i$ denotes its corresponding label. The function family $\mathcal{F}_{\mathrm{QCQP}}^{m,n}(\mathbb{R}^s)$ is constructed using the tripartite message-passing GNNs as defined in Definition 2. With all these ingredients ready, the learned function is obtained by $F = \arg\min_{f \in \mathcal{F}_{\mathrm{QCQP}}^{m,n}(\mathbb{R}^s)} \frac{1}{N} \sum_{i=1}^{N} L\left(f(G_i), y_i\right)$, where $L(\cdot, \cdot)$ is the loss function. Specifically, we use mean squared error for predicting objective values and optimal solutions, while binary cross-entropy loss is employed for predicting feasibility.

**Data generation:**  To support the supervised learning scheme mentioned above, datasets are generated by perturbing the coefficients of instances in QPLib (Furini et al., 2019). Specifically, for an arbitrary coefficient $a$ in a given instance, the coefficients of new instances are sampled from the uniform distribution $\mathcal{U}(-a, a)$. Using instances *1157*, *1493* and *1353* from

Table 1: Sizes of the base instances.

| Base ins. | # Var. | # Cons. | # Non-zeros |
|---|---|---|---|
| 1157 | 40 | 9 | 399 |
| 1493 | 40 | 5 | 240 |
| 1353 | 50 | 6 | 350 |

QPLib as base instances, we generated three datasets, each consisting of 500 instances for training and 100 ones for testing. The sizes of the base instances are listed in Table 1. To ensure convexity in these generated instances, the matrices $\{Q_i\}_{i=0}^{m}$ corresponding to the quadratic term in both the objective function and constraints are adjusted by replacing them with $Q_i - \alpha_i I$, where $\alpha_i < 0$ is the minimal eigenvalue of $Q_i$. This modification guarantees that the matrices are positive semi-definite, thereby making the corresponding QCQP instances convex. All instances are solved using the solver IPOPT solver (Wächter & Biegler, 2006) and the resulting feasibility, objective values, and optimal solutions are collected as labels.

**GNN architecture and training settings:**  For the GNN described in Section 3.1, there are three classes of functions $\{g_1^t, \ldots, g_4^t\}_{t=1}^{T}$, $\{h_1^t, \ldots, h_6^t\}_{t=1}^{T}$ and $R$ remain unspecified. The first class, $\{g_1^t, \ldots, g_4^t\}_{t=1}^{T}$, are two-layer MLPs with layer widths of $[d, d]$, and ReLU as activations, where the inputs of each function are concatenated together. The second class, $\{h_1^t, \ldots, h_6^t\}_{t=1}^{T}$, are linear transformations with output dimension $d$ followed by ReLU activations. The last one, $R$, is also a two-layer MLP with ReLU activation, with widths of $[d, 1]$ for predicting feasibility and objective values, and $[d, n]$ for predicting solutions. The hyper-parameters are set as $T = 2$ and $d = 64$. For training, we utilized the Adam optimizer alongside a one-cycle learning rate scheduler, with a maximum learning rate of 0.0001 and a batch size of 16.

**Main results:** Figure 4 illustrates the training losses for the three tasks. The subfigures from left to right correspond to the tasks of predicting feasibility, objective values, and optimal solutions, respectively. Each curve in the subfigures represents a dataset generated from one of the base instances. The results show that, for all three tasks, the training losses decrease gradually as the number of epochs increases, eventually converging to small values. Beside the curves, the best training and validation loss values during the training processes are reported in Table 2. These results validate the claim made in Theorem 1.

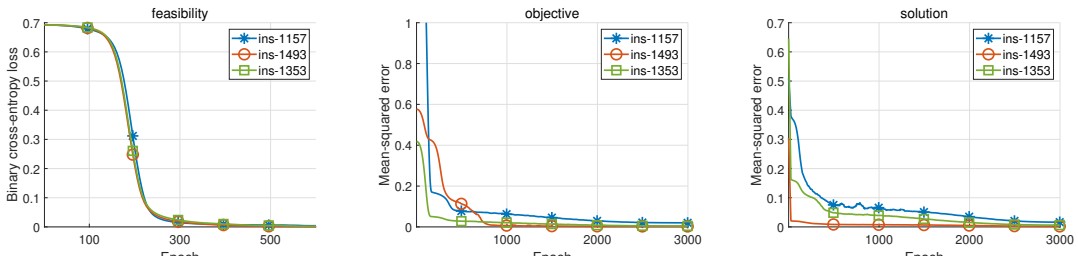

Figure 4: Training losses of predicting feasibility, objective values and optimal solutions.

Table 2: Best training and validation loss values of predicting feasibility, objective values and optimal solutions.

| Dataset | feasibility | | objective value | | optimal solution | |
|---|---|---|---|---|---|---|
| | train | validation | train | validation | train | validation |
| Perturbed from QPLIB_1157 | 1.00e-5 | 3.53e-2 | 3.67e-2 | 9.78e-2 | 4.41e-2 | 8.39e-2 |
| Perturbed from QPLIB_1493 | 4.32e-7 | 1.00e-4 | 2.05e-2 | 8.22e-2 | 9.76e-3 | 2.23e-2 |
| Perturbed from QPLIB_1353 | 2.63e-7 | 1.00e-4 | 3.42e-4 | 6.10e-3 | 6.66e-3 | 3.70e-2 |

## 5 CONCLUSIONS

This paper introduces a new tripartite graph representation specifically designed for QCQPs. By leveraging the capabilities of message-passing GNNs, this approach shows theoretical promise in predicting key properties of QCQPs with arbitrary desired accuracy, including feasibility, boundness, optimal values, and solutions. Initial numerical experiments validate the effectiveness of our framework.

This research contributes to the field of learning to optimize by expanding the application of GNNs to QCQP problems, which were previously challenging for traditional graph-based L2O methods. This could encourage future exploration in designing more specialized GNN architectures to handle QCQPs in practice, beyond the basic GCN structure employed here.

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

## A   DETAILED PROOF OF MAIN THEOREM

### A.1   SKETCH OF THE PROOF

We provide a brief outline of this complex proof:

1. **Separation Power of WL-Test:** We first establish that the WL-test has sufficient separation power on the defined target functions.

2. **Connection to tripartite message-passing GNNs:** We then demonstrate the relationship between the separation power of tripartite message-passing GNNs and that of the Tripartite WL-tests, showing that the GNNs can separate our target functions. This result, combined with the generalized Weierstrass theorem, leads to our approximation power conclusions.

3. **Universal Approximation:** Assuming the target functions are continuous and have compact support, we prove universal approximation. In this step, we also specify the problem size and apply the Generalized Weierstrass Theorem (Theorem 22 of Azizian & Lelarge (2020)).

4. **Addressing Discontinuities:** Since the target functions are neither continuous nor compactly supported, particularly at the boundary of the convex QCQPs universe $\mathcal{G}_{\mathrm{QCQP}}^{m,n}$, we construct a continuous approximation of the target function to apply universal approximation, ensuring convergence in measure.

### A.2   WL-TEST ON TRIPARTITE GRAPH REPRESENTATION

Here we describe our Tripartite WL-test, which is the WL-test counterpart of the tripartite message-passing GNNs:

- **Embedding.** Initial colors $C^{0,u}$, $C^{0,v}$, and $C^{0,c}$ are assigned based on their corresponding features and node types (e.g., from $V_1$, $V_2$, or $V_3$):
  - $C^{0,u} \leftarrow \mathrm{HASH}_1(f(u))$ for $u \in V_1$,
  - $C^{0,v} \leftarrow \mathrm{HASH}_2(f(v))$ for $v \in V_2$,
  - $C^{0,c} \leftarrow \mathrm{HASH}_3(f(c))$ for $c \in V_3$.

  Here, we refer to the color of a node after the $t$-th message-passing layer as $C^{t,\cdot}$.

- **Update quadratic nodes via variable nodes** ($V_1 \rightarrow V_2$):

$$\bar{C}^{t,v} \leftarrow \mathrm{HASH}\left(C^{t,v}, \sum_{u \in V_1} w_{u,v}\mathrm{HASH}(C^{t,u})\right), \forall v \in V_2$$

- **Update constraint nodes via variable and quadratic nodes** ($V_1, V_2 \rightarrow V_3$):

$$C^{t+1,c} \leftarrow \mathrm{HASH}\left(C^{t,c}, \sum_{u \in V_1} w_{u,c}\mathrm{HASH}(C^{t,u}), \sum_{v \in V_2} w_{v,c}\mathrm{HASH}(\bar{C}^{t,v})\right), \forall c \in V_3$$

- **Update quadratic nodes again via constraint nodes** ($V_3 \rightarrow V_2$):

$$C^{t+1,v} \leftarrow \mathrm{HASH}\left(\bar{C}^{t,v}, \sum_{c \in V_3} w_{c,v}\mathrm{HASH}(C^{t+1,c})\right), \forall v \in V_2$$

- **Update variable nodes via constraint and quadratic nodes** ($V_3, V_2 \rightarrow V_1$):

$$C^{t+1,u} \leftarrow \mathrm{HASH}\left(C^{t,u}, \sum_{c \in V_3} w_{c,u}\mathrm{HASH}(C^{t+1,c}), \sum_{v \in V_2} w_{v,u}\mathrm{HASH}(C^{t+1,v})\right), \forall u \in V_1$$

- **Termination and Readout.** Once a termination condition is met, we return the color collection $(C^{T,u})_{u \in V_1}, (C^{T,v})_{v \in V_2}, (C^{T,c})_{c \in V_3}$.[4]

- All hash functions are real-valued and assumed to be collision-free.

In this paper, we terminate the Tripartite WL-test only when the algorithm stabilizes[5], i.e., when the number of distinct colors no longer changes in an iteration (after all four color updates). Despite not imposing a forced iteration limit, the WL-test is guaranteed to terminate in a finite number of iterations, denoted by $T$:

**Proposition 3** (Tripartite WL-test terminates in finite iterations). *The Tripartite WL-test stabilizes in a finite number of iterations.*

*Proof.* It is straightforward to observe from the formulation that if two nodes have different colors, they will continue to have different colors after an (sub-)iteration. Therefore, the number of iterations required for stabilization is capped by the number of distinct nodes, which is finite. □

We say that the Tripartite WL-test *separates* two graphs if the resulting collection of colors differs between the two graphs. We claim that the Tripartite WL-test has the same separation power as its network counterpart, specifically the tripartite message-passing GNNs:

**Proposition 4** (tripartite message-passing GNNs have equal separation power as the Tripartite WL-Test). *Given two instances $\mathcal{I}$ and $\bar{\mathcal{I}}$ (correspondingly encoded by graphs $G$ and $\bar{G}$), the following holds:*

1. *For graph-level output cases, the two instances are separated by $\mathcal{F}_{\mathrm{QCQP}}^{m,n}(\mathbb{R})$, i.e.,*

$$F(G) = F(\bar{G}), \forall F \in \mathcal{F}_{\mathrm{QCQP}}^{m,n}(\mathbb{R})$$

   *if and only if the two instances are also separated by the Tripartite WL-test.*

2. *For node-level output cases, i.e., $\mathbb{R}^s = \mathbb{R}^n$, the two instances are separated by $\mathcal{F}_{\mathrm{QCQP}}^{m,n}(\mathbb{R})$, i.e.,*

$$F(G) = F(\bar{G}), \forall F \in \mathcal{F}_{\mathrm{QCQP}}^{m,n}(\mathbb{R}^n)$$

   *if and only if the two instances are separated by the Tripartite WL-test, and additionally, the variables are correspondingly indexed. Specifically, $C^{T,u_j} = C^{T,\bar{u}_j}$ must hold for all $j \in [n]$.*

For the detailed proof of this proposition, see Appendix B.3.

### A.3 PROOF OF MAIN THEOREM

Now we can prove the main theorem. First, we state our key lemma:

**Lemma 1.** *Let $\mathcal{I}, \bar{\mathcal{I}}$ (with given sizes $m, n$, encoded by $G, \bar{G} \in \mathcal{G}_{\mathrm{QCQP}}^{m,n}$) be two QCQP instances. If the following holds:*

- *The tripartite WL-test cannot separate the two instances;*

- *$x$ is a feasible solution of $\mathcal{I}$.*

*Then there exists a feasible solution $\bar{x}$ for $\bar{\mathcal{I}}$ whose objective and $\ell_2$-norm are controlled by $x$, such that:*

$$\bar{x}^\top \bar{Q} \bar{x} + \bar{p} \cdot \bar{x} \leq x^\top Q x + p \cdot x$$
$$||\bar{x}|| \leq ||x||$$

---

[4]Multiple occurrences of members are counted instead of rejected.

[5]For simplicity, we exclude the final iteration showing that the algorithm has stabilized and return the last iteration in which stabilization occurred.

For the detailed proof of this lemma, see Appendix B.2. With this key lemma, we derive the following corollary, which establishes the separation power of the Tripartite WL-test and tripartite message-passing GNNs, since they have equal separation power.

**Proposition 5.** *Let $\mathcal{I}, \bar{\mathcal{I}}$ (encoded by $G, \bar{G} \in \mathcal{G}_{\text{QCQP}}^{m,n}$) be two QCQP instances. If the tripartite WL-test fails to separate the two instances, then the following holds:*

1. *If one is feasible, the other is also feasible, i.e., $\Phi_{\text{feas}}(G) = \Phi_{\text{feas}}(\bar{G})$.*

2. *Assume both instances are feasible. If one is unbounded, the other is also unbounded.*

3. *Assume both instances are bounded. Then they have equal optimal values, i.e., $\Phi_{\text{obj}}(G) = \Phi_{\text{obj}}(\bar{G})$.*

4. *Assume both instances are bounded and that the variables and constraints are indexed such that $C^{T,u_j} = C^{T,\bar{u}_j}$. Then they have the same optimal solution, with the least $L^2$-norm, i.e., $\Phi_{\text{sol}}(G) = \Phi_{\text{sol}}(\bar{G})$.*

*Proof.* **Passing feasibility**. Assume that $\mathcal{I}$ is feasible, and let $x$ be a feasible solution. By Lemma 1, we obtain another solution $\bar{x}$ for instance $\bar{\mathcal{I}}$, which implies the feasibility of $\bar{\mathcal{I}}$. By switching the roles of $\mathcal{I}$ and $\bar{\mathcal{I}}$, we prove the reverse claim.

**Passing unboundedness**. Assume that $\mathcal{I}$ is unbounded, i.e., for any $M > 0$, there exists a solution $x_M$ such that the objective $f(x) \le -M$. For each $x_M$, we can construct a solution $\bar{x}_M$ for $\bar{\mathcal{I}}$ such that the objective $\bar{f}(\bar{x}_M) \le f(x_M) \le -M$, implying that $\bar{\mathcal{I}}$ is also unbounded. Again, by switching the roles of $\mathcal{I}$ and $\bar{\mathcal{I}}$, we prove the reverse claim.

**Passing optimal value**. Assume that $\mathcal{I}$ is feasible and bounded, and let $x$ be its optimal solution. By Lemma 1, we construct a solution $\bar{x}$ for $\bar{\mathcal{I}}$ such that:

$$\bar{f}(\bar{x}) \le f(x) = \Phi_{\text{obj}}(G)$$

implying that $\Phi_{\text{obj}}(\bar{G}) \le \Phi_{\text{obj}}(G)$. Similarly, we can show that $\Phi_{\text{obj}}(G) \le \Phi_{\text{obj}}(\bar{G})$, and thus $\Phi_{\text{obj}}(\bar{G}) = \Phi_{\text{obj}}(G)$.

**Passing optimal solution**. To prove the last claim, we need the construction of $\bar{x}$ from the detailed proof of Lemma 1 (see Appendix B.2). Assume that $\mathcal{I}$ is feasible and bounded, and let $x$ be its optimal solution (with the least $L^2$-norm). By Lemma 1, we construct $y$ for $\bar{\mathcal{I}}$ and $z$ for $\mathcal{I}$ by switching the roles of $\mathcal{I}$ and $\bar{\mathcal{I}}$.

We have $f(z) \le \bar{f}(y) \le f(x)$ and $\|z\| \le \|y\| \le \|x\|$, which implies that $z$ is not worse than the given optimal solution $x$, and thus $z = x$. By the construction of the averaged solution (and the assumption $C^{T,u_j} = C^{T,\bar{u}_j}$), we have $y = z$. Combining the two equalities, we conclude that $x = y$.

Let $\bar{x}$ be the optimal solution of $\bar{G}$, and we have $\|\bar{x}\| \le \|y\| = \|x\|$. By switching the roles of $\mathcal{I}$ and $\bar{\mathcal{I}}$, we obtain $\|x\| \le \|\bar{x}\|$, and thus $\|\bar{x}\| = \|x\|$. Similarly, we have $\bar{f}(\bar{x}) \le \bar{f}(y) \le f(x)$, and by switching the roles, $\bar{f}(\bar{x}) = f(x)$.

Since $\|y\| = \|x\| = \|\bar{x}\|$ and $\bar{f}(y) = f(x) = f(\bar{x})$, by uniqueness, we conclude that $y = \bar{x}$, proving the fourth claim. $\square$

The next step is to extend this separation power to approximation power, which leads to our main theorem. We utilize the generalized Weierstrass-Stone theorem (Theorem 22 and Lemma 36 of Azizian & Lelarge (2020)) and Lusin's theorem.

By applying the generalized Weierstrass-Stone theorem, we establish the following proposition, which demonstrates the approximation power on equivariant functions with compact support:

**Proposition 6** (Uniform Approximation on Continuous Equivariant Functions with Compact Support). *Let $\Phi_c : \mathcal{G}_c^{m,n} \to \mathbb{R}^s$ be a general continuous target function defined on a compact subset $\mathcal{G}_c \subseteq \mathcal{G}_{\mathrm{QCQP}}^{m,n}$, such that:*

- *If $s = 1$, the output remains unchanged if the input graph is re-indexed.*

- *If $s = n$, the output re-indexes accordingly if the input graph is re-indexed.*

*If the following holds:*

$$\left( F(G) = F(\bar{G}), \forall F \in \mathcal{F}_{\mathrm{QCQP}}^{m,n}(\mathbb{R}^s) \Rightarrow \Phi(G) = \Phi(\bar{G}) \right), \forall G, \bar{G} \in \mathcal{G}_c^{m,n} \tag{A.1}$$

*i.e., the family $\mathcal{F}_{\mathrm{QCQP}}^{m,n}(\mathbb{R}^s)$ separates the target function $\Phi$, then for any $\delta > 0$, there exists a function $F_\delta \in \mathcal{F}_{\mathrm{QCQP}}^{m,n}(\mathbb{R}^s)$ such that:*

$$\|F_\delta(\mathcal{G}) - \Phi(G)\| < \delta \tag{A.2}$$

For the detailed proof, see Appendix B.4.

However, the requirement for the target function to apply the proposition is too strong. In fact, all target functions defined in 3 are non-continuous and not defined on a compact subset, although equivariance naturally holds. Therefore, we seek a continuous approximation with compact support that can be uniformly approximated. By applying Lusin's theorem, we construct the following continuous approximation:

**Proposition 7** (Continuous Approximation with Compact Support). *Let $\Phi : \mathcal{G}_{\mathrm{QCQP}}^{m,n} \to \mathbb{R}^s$ be a general target function that is measurable under the probability measure $\mathbb{P}$. For any $\varepsilon > 0$, there exists a compact subset $\mathcal{G}_c^{m,n} \subseteq \mathcal{G}_{\mathrm{QCQP}}^{m,n}$, such that $\mathbb{P}\{G \in \mathcal{G}_c^{m,n}\} > 1 - \varepsilon$, and $\Phi|_{\mathcal{G}_c^{m,n}}$ is continuous.*

By combining all the lemmas and propositions, we can now prove the main theorem.

*Proof of Theorem 1.* Let $\Phi$ be any target function defined in Definition 3.

By Proposition 7, $\Phi$ is continuous on a compact subset $\mathcal{G}_c^{m,n} \subseteq \mathcal{G}_{\mathrm{QCQP}}^{m,n}$, with $\mathbb{P}(G \in \mathcal{G}_c^{m,n}) \geq 1 - \frac{\varepsilon}{|\Sigma|}$.

We construct $\mathcal{G}_{c,\mathrm{eq}}^{m,n} = \cap_{(\sigma,\tau) \in \Sigma}(\sigma, \tau)(\mathcal{G}^{m,n})$. This subset is continuous with compact support, ensuring that $\Phi|_{\mathcal{G}_{c,\mathrm{eq}}^{m,n}}$ remains an equivariant function, with the following measure control:

$$\mathbb{P}(G \in \mathcal{G}_{c,\mathrm{eq}}^{m,n}) > 1 - \varepsilon \tag{A.3}$$

Since by Proposition 5 and the fact that the Tripartite WL-test has equal separation power as the tripartite message-passing GNNs, the target functions are equivariant and separated by $\mathcal{F}_{\mathrm{QCQP}}^{m,n}(\mathbb{R}^s)$. Thus, we may apply Proposition 6 and obtain $F \in \mathcal{F}_{\mathrm{QCQP}}^{m,n}(\mathbb{R}^s)$ such that:

$$\|F(G) - \Phi(G)\| < \delta, \forall G \in \mathcal{G}_{c,\mathrm{eq}}^{m,n}$$

This implies that $\mathbb{P}\{\|F(G) - \Phi(G)\| < \delta\} > 1 - \varepsilon$. $\qquad\qquad\square$

# B  PROOF OF PROPOSITIONS IN SECTION  A

This section provides complete proofs of several propositions in Section A that were not immediately proven.

## B.1 EQUIVARIANCE

We begin by describing equivariance, a key tool used to capture the fact that the indexing of variables and constraints is irrelevant:

**Definition 4.** *Given a function $f : X \to Y$, where $X$ and $Y$ are subsets of Euclidean spaces, and a group $\Sigma$ that acts continuously on $X$ and $Y$, the function $f$ is called equivariant (with respect to the group $\Sigma$) if the following holds:*

$$\sigma \circ f(x) = f \circ \sigma(x), \quad \forall x \in X, \sigma \in \Sigma$$

Since the indexing of variables and constraints does not affect the problem, we take $\Sigma = S_n \times S_m$, which represents all possible re-indexings of variables and constraints. When applied to both the input and output spaces, we re-index the variables, constraints, and possible solutions (in cases where the output is a solution $x \in \mathbb{R}^n$). Specifically, we have:

$$\tilde{q}_{\pi(j),\pi(k)} = q_{j,k}$$
$$\tilde{p}_{\pi(j)} = p_j$$
$$\tilde{q}^{\tau(i)}_{\pi(j),\pi(k)} = q^i_{j,k}$$
$$\tilde{p}^{\tau(i)}_{\pi(j)} = p^i_j$$
$$\tilde{b}_{\tau(i)} = b_i$$
$$\tilde{x}^{\mathrm{L}}_{\pi(j)} = x^{\mathrm{L}}_j$$
$$\tilde{x}^{\mathrm{U}}_{\pi(j)} = x^{\mathrm{U}}_j$$

where the tilde symbols $\tilde{Q}, \tilde{p}, \tilde{b}, \tilde{x}^{\mathrm{L}}, \tilde{x}^{\mathrm{U}}$ denote the re-indexed vectors and matrices.

For $\mathbb{R}^s = \mathbb{R}$, the action on the output space is the identity map: $(\pi, \tau)(\cdot) = \mathrm{id}$. For $\mathbb{R}^s = \mathbb{R}^n$, we correspondingly re-index the output, i.e., $(\pi, \tau)(y)_{\pi(j)} = y_j$.

We can also apply the permutations to:

- A point in $\mathbb{R}^n$ (such as a solution), by $(\pi, \tau)(x)_{\pi(j)} = x_j$.
- A subset of $\mathbb{R}^n$, by applying the permutation to each element in the subset, or to its indicator function by permuting the underlying set.

Equivariance allows us to show that the indices do not matter, while the inputs (in the form of coefficient tuples) necessarily carry these indices.

**Remark**: Given the group $\Sigma$ and its action on both the input and output, all message-passing layers are automatically equivariant. Thus, requiring $F \in \mathcal{F}^{m,n}_{\mathrm{QCQP}}(\mathbb{R}^s)$ to be equivariant is equivalent to requiring the readout layer $R$ to be equivariant. This is why the readout function must take specific forms in the two cases. While the defined forms do not cover all possible equivariant readout functions, they are general enough to capture the separation power.

## B.2 PROOF OF CORE LEMMA

For simplicity of proof, we extend the definitions of $\Phi_{\mathrm{obj}}$ and $\Phi_{\mathrm{sol}}$ to the entire space $\mathcal{G}^{m,n}_{\mathrm{QCQP}}$ by assigning a default value of $0$ (or $\mathbf{0}$, depending on the output dimension $s$) when the target function is not defined at a graph $G$. This occurs when the corresponding instance is either infeasible or unbounded, and the optimal value or optimal solution does not exist. By doing so, all target functions are defined on the same space $\mathcal{G}^{m,n}_{\mathrm{QCQP}}$. Moreover, since we approximate feasibility and boundedness, we can distinguish whether the output is the default value or genuinely happens to be $0$ (or $\mathbf{0}$).

Let $\mathcal{I}$ and $\bar{\mathcal{I}}$ be two instances (with Tripartite graph representations $G$ and $\bar{G} \subseteq \mathcal{G}_{\text{QCQP}}^{m,n}$) that are not separated by the Tripartite WL-test. Without loss of generality, we assume that the variables and constraints are correspondingly indexed, i.e., $C^{T,u_j} = C^{T,\bar{u}_j}$ and $C^{T,c_i} = C^{T,\bar{c}_i}$ hold for all $i, j$.

We first introduce the following notations. Let $I$ be any color, and we collect all nodes of a graph $G$ with color $I$, denoting this collection as $G(I)$. Throughout this paper, we use $J$ for the colors of variable nodes, $K$ for quadratic nodes, and $I$ for constraint nodes.

We now present the following lemma:

**Lemma 2.** *Given the graph $G$, let the Tripartite WL-test stabilize after $T \geq 0$ iterations. The sum of weights from a certain node of one color to all nodes of another color depends only on the color of the given node. Specifically, the sum (taking $J$ for variable nodes and $K$ for quadratic nodes as an example) is:*

$$S(J, K; G) := \sum_{C^{T,v}=K} w_{u,v}$$

*and is well-defined with $u \in G(J)$ arbitrarily chosen.*

*Similarly, for any color of constraints $I$, color of variables $J$, and color of quadratic terms $K$, the following sums are well-defined:*

$$S(J, I; G) := \sum_{C^{T,c}=I} w_{u,c}, \quad C^{T,u} = J$$

$$S(I, K; G) := \sum_{C^{T,v}=K} w_{c,v}, \quad C^{T,c} = I$$

$$S(K, I; G) := \sum_{C^{T,c}=I} w_{v,c}, \quad C^{T,v} = K$$

$$S(J, K; G) := \sum_{C^{T,v}=K} w_{u,v}, \quad C^{T,u} = J$$

$$S(K, J; G) := \sum_{C^{T,u}=J} w_{v,u}, \quad C^{T,v} = K$$

*Proof.* Let $v, v'$ be two nodes with color $K = C^{T,v} = C^{T,v'}$. Since the Tripartite WL-test has stabilized, further iterations do not separate additional node pairs, i.e.,

$$\sum_u w_{u,v}\text{HASH}(C^{T,u}) = \sum_u w_{u,v'}\text{HASH}(C^{T,u}).$$

Rearranging according to $J = C^{T,u}$, we get:

$$\sum_J \sum_{C^{T,u}=J} w_{u,v} \cdot \text{HASH}(J) = \sum_J \sum_{C^{T,u}=J} w_{u,v'} \cdot \text{HASH}(J).$$

Assuming that the hash function is collision-free, we conclude that:

$$\sum_{C^{T,u}=J} w_{u,v} = \sum_{C^{T,u}=J} w_{u,v'},$$

i.e., $S(K, J; G) := \sum_{C^{T,u}=J} w_{v,u}, \quad C^{T,v} = K$ is well-defined.

The other claims follow similarly. $\qquad\square$

By summing all weights between two colors $I$ and $J$, we derive the following lemma:

**Lemma 3.** *Let $J$ and $K$ be arbitrary node colors. Then, the following holds:*

$$|G(J)|S(J, K; G) = |G(K)|S(K, J; G),$$

*and similar equalities hold between $I$ and $J$, and between $I$ and $K$.*

*Proof.* Summing all edges between all nodes with $C^{T,u} = J$ and $C^{T,v} = K$, and re-arranging the sum according to $u$ and $v$, by Lemma 2, we have:

$$|G(J)|S(J, K; G) = |G(K)|S(K, J; G).$$

The other two claims are similar. □

We are now ready to proceed. We construct $\bar{x}_j = \frac{1}{|G(J)|} \sum_{j':C^{T,u}{}_{j'}=C^{T,\bar{u}}{}_{j'}=J} x_{j'}$, where $J = C^{T,x_j}$. We claim that $\bar{x}$ satisfies all the required conditions.

First, we analyze the **linear part** of the constraints and the objective. Let $f^i_{\mathrm{lin}}(x) := p^i \cdot x$ represent the linear part of the $i$-th constraint. For a certain color $I$ of constraint nodes, we have:

$$
\begin{aligned}
\bar{f}^i_{\mathrm{lin}}(\bar{x}) &= \sum_j \bar{p}^i_j \bar{x}_j \\
&= \sum_J \sum_{\bar{v}_j \in G(J)} \bar{p}^i_j \bar{x}_j \\
&= \sum_J S(I, J) \bar{x}_J \\
&= \frac{1}{|G(I)|} \sum_J S(J, I) |G(J)| \bar{x}_J \\
&= \frac{1}{|G(I)|} \sum_J S(J, I) \sum_{u_j \in G(J)} x_j \\
&= \frac{1}{|G(I)|} \sum_{c_i \in G(I)} \sum_J \sum_{j \in G(J)} p^i_j x_j \\
&= \frac{1}{|G(I)|} \sum_{c_i \in G(I)} f^i_{\mathrm{lin}}(x).
\end{aligned}
$$

(B.1)

Here, $\bar{x}_j$ is the average over the nodes with color $J$, so it is determined by $J$, and we denote its value as $\bar{x}_J$.

We define $f_{\mathrm{lin}}(x) = p \cdot x$. For the objective part, we have:

$$
\begin{aligned}
\sum_j \bar{p}_j \bar{x}_j &= \sum_J p_J |\bar{G}(J)| \bar{x}_J \\
&= \sum_J p_J \sum_{u_j \in G(J)} x_j \\
&= \sum_j p_j x_j,
\end{aligned}
$$

(B.2)

where $p_j, \bar{p}_j$ are the features of the variables, which are determined by the color $J = C^{T,u_j} = C^{T,\bar{u}_j}$. We denote this value by $p_j = \bar{p}_j = p_J$.

**Quadratic part**. We define $f^i_{\mathrm{quad}}(x) = \frac{1}{2} x^\top q^i x$ as the **quadratic** part of the $i$-th constraint.

For a certain color $I$ of constraint nodes, we have the following:

$$\bar{f}^i_{\text{quad}}(\bar{x}) = \frac{1}{2} \sum_{v_{j,k} \in V_2(\bar{G})} \bar{q}^i_{j,k} \bar{x}_j \bar{x}_k$$

$$= \frac{1}{2} \sum_K \sum_{\bar{v}_{j,k} \in \bar{G}(K)} \bar{q}^i_{j,k} \bar{x}_j \bar{x}_k$$

$$= \frac{1}{2} \sum_K S(I,K) |\bar{G}(K)| \bar{x}_K.$$

Since all $\bar{v}_{j,k} \in V_2(\bar{G})$ have $\bar{u}_j, \bar{u}_k$ as neighbors in $V_1(\bar{G})$, $\bar{x}_K := \bar{x}_j \bar{x}_k$ is well-defined. This equation shows that the value $\bar{f}^i_{\text{quad}}(\bar{x})$ depends only on the color $I = C^{T, \bar{c}_i}$, and not on the specific selection of $\bar{c}_i \in \bar{G}(I)$. Therefore, $f^i_{\text{quad}}(\bar{x})$ reduces to the sum, and we claim that $f^i_{\text{quad}}(\bar{x}) = \bar{f}^i_{\text{quad}}(\bar{x})$ holds.

Next, we consider the partial derivative. Let $J := C^{T, u_j}$, and we have:

$$\partial_j \sum_{c_i \in G(I)} f^i_{\text{quad}}(\bar{x}) = \sum_{c_i \in G(I)} \sum_k w(u_j, v_{j,k}) w(v_{j,k}, c_i) \bar{x}_k$$

$$= \sum_{c_i \in G(I)} \sum_K \sum_{k: v_{j,k} \in G(K)} w(u_j, v_{j,k}) w(v_{j,k}, c_i) \bar{x}_k \quad \text{(B.3)}$$

$$= \sum_K S(K, I) \sum_{k: v_{j,k} \in G(K)} w(u_j, v_{j,k}) \bar{x}_k$$

$$= \sum_K S(K, I) S(J, K) x_{K;J}.$$

Since $u_j$ is one of the neighbors in $v_{j,k}$, and $v_{j,k} \in G(K)$ has exactly two neighbors in $V_1(G)$, we know that the color of $u_k$ depends only on the colors $K = C^{T, v_{j,k}}$ and $J = C^{T, v_j}$. This makes $\bar{x}_{K;J} := \bar{x}_k$ well-defined, with $u_j \in G(J)$ and $v_{j,k} \in G(K)$.

Thus, the derivative $\partial_j \sum_{c_i \in G(I)}$ depends only on $J = C^{T, u_j}$, i.e.,

$$C^{T, v_{j_1}} = C^{T, v_{j_2}} \Rightarrow \partial_{j_1} \sum_{c_i \in G(I)} f^i_{\text{quad}}(\bar{x}) = \partial_{j_2} \sum_{c_i \in G(I)} f^i_{\text{quad}}(\bar{x}). \quad \text{(B.4)}$$

By Equation equation B.4, we know that $\bar{x}$ is a local optimal point within the linear space:

$$\{y \in \mathbb{R}^n : \sum_{u_j \in G, C^{T, u_j} = J} y_j = \sum_{u_j \in G, C^{T, u_j} = J} x_j\}.$$

With the convexity assumption, the local optimal point is a global minimum. Since $x$ is in this linear space, we claim that:

$$\sum_{c_i \in G(I)} f^i_{\text{quad}}(\bar{x}) \leq \sum_{c_i \in G(I)} f^i_{\text{quad}}(x). \quad \text{(B.5)}$$

Combining Equation equation B.5 with the fact that $f^i_{\text{quad}}(\bar{x})$ and $\bar{f}^i_{\text{quad}}(\bar{x})$ are equal for all $c_i \in G(I)$, we can control the quadratic parts:

$$\bar{f}^i_{\text{quad}}(\bar{x}) = f^i_{\text{quad}}(\bar{x})$$

$$= \frac{1}{|G(I)|} \sum_{c_i \in G(I)} f^i_{\text{quad}}(\bar{x}) \quad \text{(B.6)}$$

$$\leq \frac{1}{|G(I)|} \sum_{c_i \in G(I)} f^i_{\text{quad}}(x).$$

For the objective part, we define $f_{\text{quad}}(x) = \frac{1}{2}x^\top Q x$. Similarly, we have:

$$\bar{f}_{\text{quad}}(\bar{x}) = \frac{1}{2}\sum_{\bar{v}_{j,k}} f^0(\bar{v}_{j,k})\bar{x}_j\bar{x}_k$$

$$= \frac{1}{2}\sum_K \sum_{\bar{v}_{j,k}\in\bar{G}(K)} f^0(\bar{v}_{j,k})\bar{x}_j\bar{x}_k$$

$$= \frac{1}{2}\sum_K \sum_{v_{j,k}\in G(K)} f^0(v_{j,k})\bar{x}_j\bar{x}_k$$

$$= f_{\text{quad}}(\bar{x}).$$

We also have:

$$\partial_j f_{\text{quad}}(\bar{x}) = \sum_k f^0(v_{j,k})w(u_j, v_{j,k})\bar{x}_k$$

$$= \sum_K \sum_{k:v_{j,k}\in G(K)} f^0(v_{j,k})w(u_j, v_{j,k})\bar{x}_k$$

$$= \sum_K f^0(K)S(J,K)\bar{x}_{K;J},$$

which depends only on $J = C^{T,u_j}$. Here, $f^0(K) = f^0(v_{j,k})$, and $v_{j,k} \in G(K)$ is well-defined by the stable color assumption.

**Combination of the two parts.**

The color $C^{T,c_i} = C^{T,\bar{c}_i} = I$ determines the RHS $b_I := b_i$. Defining $f_{\text{cons}}^i(x) = f_{\text{quad}}^i(x) + f_{\text{lin}}^i(x)$, and similarly for $\bar{\mathcal{I}}$, we have:

$$\bar{f}_{\text{cons}}^i(\bar{x}) = \bar{f}_{\text{quad}}^i(\bar{x}) + \bar{f}_{\text{lin}}^i(\bar{x})$$

$$= \frac{1}{|G(I)|}\sum_{c_i\in G(I)}\left(f_{\text{quad}}^i(x) + f_{\text{lin}}^i(x)\right)$$

$$= \frac{1}{|G(I)|}\sum_{c_i\in G(I)} f_{\text{cons}}^i(x)$$

$$\leq b_I.$$

For the objective, we similarly have:

$$\bar{f}_{\text{quad}}(\bar{x}) + \bar{f}_{\text{lin}}(\bar{x}) \leq f_{\text{quad}}(x) + f_{\text{lin}}(x).$$

This completes the proof that $\bar{x}$ is the solution for $\bar{\mathcal{I}}$, satisfying the condition given in Proposition 1.

### B.3 PROOF OF PROPOSITION 4

We prove the separation power by simulating the tripartite WL-test using tripartite message-passing GNNs. We define the hidden representation $h^{t,\cdot}$, produced by some network, as a **one-hot representation** of the colors $C^{t,\cdot}$ if all $h^{t,\cdot}$ are one-hot vectors, and they take the same value if and only if they have the same color $C^{t,\cdot}$.

First, we consider the color initialization. We collect all the features paired with the node types (i.e., variable nodes, quadratic nodes, and constraint nodes). Then we select $g_{1,2,3}^0$ to map the features to

one-hot vectors, where the enumeration serves as the only index with the value $1.0$. For example, if the feature $h^{u_j}$ of a variable node is enumerated by $r$, then $g_1^0$ maps $h^{u_j}$ to $h^{0,u_j} = e_r$.

It's easy to see that the embedded hidden feature $h^{0,\cdot}$ is a one-hot representation of the initial color $C^{0,\cdot}$.

Next, we consider the first refinement. Assuming that $h^{t,\cdot}$ is a one-hot representation of $C^{t,\cdot}$ and $g_1^t = \mathrm{id}$ is a simple and proper hash function, the concatenated vector

$$\left[ h^{t,v}, \sum_{u \in V_1} w_{u,v} f_1^t(h^{t,u}) \right]$$

is a representation of the colors $\bar{C}^{t,\cdot}$, which is generally not one-hot. The same holds for the other three concatenated vectors from the remaining three sub-layers. By Theorem 3.2 of Yun et al., 2019, a network with four fully connected layers and ReLU activation maps these values back to one-hot. Therefore, we select $f_1^t$ to concatenate the inputs and then pass them through a 4-layered MLP with ReLU activation, so that the aggregated hidden representation $\bar{h}^{t,\cdot}$ is once again one-hot.

Similarly, we get $h_{2,3,4}^t$ and $g_{2,3,4,5,6}^t$ and simulate an iteration of the Tripartite WL-test with a round of four message-passing sub-layers.

In the case of graph-level output, the readout function takes the following form:

$$R(\cdot) = f_{\text{out}} \left( \sum_j h^{T,u_j}, \sum_{j,k} h^{T,v_{j,k}}, \sum_i h^{T,c_i} \right).$$

Since the hidden representation is a one-hot representation of $C^{T,\cdot}$, if two instances are not separated by the tripartite message-passing GNN, they are not separated by this subset of GNNs (given a fixed initialization and a free readout function). Consequently, all entries must be equal, and the two instances are not separated by the Tripartite WL-test.

Similarly, in the case of node-level output, all equivariant readout functions take the form:

$$R(\cdot)_j = f_{\text{out}} \left( h^{T,u_j}, \sum_j h^{T,u_j}, \sum_{j,k} h^{T,v_{j,k}}, \sum_i h^{T,c_i} \right).$$

Thus, all entries must be equal, and the two instances are not separated. Moreover, the variables are correspondingly indexed.

Conversely, we use induction to prove that for all $t \in \mathbb{N}$, the colors $C^{t,\cdot}$ separate more than the hidden features $h^{t,\cdot}$, i.e.,

$$C^{t,u} = C^{t,u'} \Rightarrow h^{t,u} = h^{t,u'}, \quad \forall u, u' \in V_1 \cup \bar{V}_1, F \in \mathcal{F}_{\text{QCQP}}^{m,n}(\mathbb{R}^s), \tag{B.7}$$

and similar claims hold for the other three sub-iterations.

For $t = 0$ (i.e., right after embedding), the statement is obviously true. Now, assume that after some sub-iteration (say, before the first sub-iteration of iteration $t \geq 1$, with the other sub-iterations following similarly), the statement holds.

Let $v, v'$ satisfy:

$$\sum_u w_{u,v} \mathrm{HASH}(C^{t,u}) = \sum_u w_{u,v'} \mathrm{HASH}(C^{t,u}).$$

Organizing the sum by $C^{t,u} = J$, and assuming the hash function is collision-free, we have:

$$\sum_{u:C^{t,u}=J} w_{u,v} = \sum_{u:C^{t,u}=J} w_{u,v'}, \quad \forall J. \tag{B.8}$$

Next, we organize the sum $\sum_u w_{u,v} f_1^t(h^{t,u})$ by the value of $h^{t,u}$. By the induction assumption, the set $\{u : h^{t,u} = h\}$ is the union of $\{u : C^{t,u} = J_l\}$ for some colors $J_l$. Summing the equality in equation B.8 over the colors, we have:

$$\sum_{h^{t,u}=h} w_{u,v} = \sum_{h^{t,u}=h} w_{u,v'}, \quad \forall h.$$

Thus, we conclude:

$$\sum_u w_{u,v} f_1^t(h^{t,u}) = \sum_h \sum_{u:h^{t,u}=h} w_{u,v} f_1^t(h) = \sum_h \sum_{u:h^{t,u}=h} w_{u,v'} f_1^t(h) = \sum_u w_{u,v'} f_1^t(h^{t,u}),$$

which completes the induction.

For the case of graph-level output, this means that all entries of the input to the readout function are equal for the two graphs, i.e.,

$$\left( \sum_j h^{T,u_j}, \sum_{j,k} h^{T,v_{j,k}}, \sum_i h^{T,c_i} \right) = \left( \sum_j h^{T,\bar{u}_j}, \sum_{j,k} h^{T,\bar{v}_{j,k}}, \sum_i h^{T,\bar{c}_i} \right),$$

and the GNNs give the same output for all possible readout functions.

For the case of node-level output, we again have:

$$\left( h^{T,u_j}, \sum_j h^{T,u_j}, \sum_{j,k} h^{T,v_{j,k}}, \sum_i h^{T,c_i} \right) = \left( h^{T,\bar{u}_j}, \sum_j h^{T,\bar{u}_j}, \sum_{j,k} h^{T,\bar{v}_{j,k}}, \sum_i h^{T,\bar{c}_i} \right).$$

Here, we use the assumption that the variables are correspondingly indexed to guarantee $h^{T,u_j} = h^{T,\bar{v}_j}$.

### B.4 PROOF OF PROPOSITION 6

The requirement for the general target function $\Phi_c$ is simply equivariance under re-indexing. Thus, we need to verify the conditions required by the generalized Weierstrass theorem (Theorem 22 of Azizian & Lelarge (2020)) to apply.

First, we verify that $\mathcal{F} = \mathcal{F}_{\text{QCQP}}^{m,n}(\mathbb{R}^s)$ is a sub-algebra. By multiplying the readout function by $\lambda$, we construct $\lambda F \in \mathcal{F}_{\text{QCQP}}^{m,n}(\mathbb{R}^s)$. Now, we construct the sum and product of two functions $F_1, F_2 \in \mathcal{F}_{\text{QCQP}}^{m,n}(\mathbb{R}^s)$.

Given $F_1$ and $F_2$, we proceed as follows:

- We construct
$$g_{1,F}^0(h^{0,u}) := \left[ g_{1,F_1}^0(h^{0,u}), g_{1,F_2}^0(h^{0,u}) \right].$$
We give similar constructions for $g_{2,F}^0$ and $g_3^0$.

- After initialization, all hidden features take the form $h^{t,u} = [h_{F_1}^{t,u}, h_{F_2}^{t,u}]$ (considering variable nodes as an example, and similarly for quadratic nodes). We construct
$$g_{1,F}^t(h^{t,u}) := \left[ g_{1,F_1}^t(h_{F_1}^{t,u}), g_{1,F_2}^t(h_{F_2}^{t,u}) \right],$$
and
$$f_{1,F}^t\left(h^{t,v}, \sum_u w_{uv} h^{t,u}\right) := \left[ f_{1,F_1}^t\left(h_{F_1}^{t,v}, \sum_u w_{uv} h_{F_1}^{t,u}\right), f_{1,F_2}^t\left(h_{F_2}^{t,v}, \sum_u w_{uv} h_{F_2}^{t,u}\right) \right].$$
We give similar constructions for other $g_{\cdot,F}^t, f_{\cdot,F}^t$. Using this construction, we compute both hidden representations in one concatenated network.

- Finally, we obtain $F = F_1 + F_2$ by constructing $R(\cdot) = R_1(\cdot_{F_1}) + R_2(\cdot_{F_2})$, and similarly for $F = F_1 \times F_2$.

Thus, we conclude that $F_1 + F_2, F_1 \times F_2 \in \mathcal{F}_{\text{QCQP}}^{m,n}(\mathbb{R}^s)$.

Next, we verify the inclusion $\rho(\mathcal{F}_{\text{scal}}) \subseteq \rho(\pi_\Sigma \circ \mathcal{F})$:

**Graph-level output case**. In this case, we have $\mathcal{F}_{\text{scal}} = \mathcal{F}$ and $\pi_\Sigma = \text{id}$, so the two sides are exactly the same.

**Node-level output case**. Given any $R_1$ that maps the final hidden representation to a graph-level output, $R_1 \cdot \mathbf{1}_n = (R_1, R_1, \ldots, R_1)$ is a valid equivariant readout function in the node-level case. Thus, given any $F \in \mathcal{F}_{\text{QCQP}}^{m,n}(\mathbb{R})$, we can construct $F' \in \mathcal{F}_{\text{QCQP}}^{m,n}(\mathbb{R}^n)$ using $R_1 \cdot \mathbf{1}_n$, along with all the $f$ and $g$ functions, and conclude that any pair $(G_1, G_2) \in \rho(\mathcal{F}_{\text{scal}})$ is not separated by the Tripartite WL-test.

For any pair of graphs $(G, \bar{G})$ that is not separated by the Tripartite WL-test, after re-indexing variables and constraints, all $F \in \mathcal{F}$ map them to the same output. This means that, without re-indexing, all $F \in \mathcal{F}$ map the two graphs to outputs that differ at most by a re-indexing. Thus, $(G, \bar{G})$ is contained in $\rho(\pi_\Sigma \circ \mathcal{F})$. This completes our verification.

Applying the Generalized Weierstrass-Stone theorem to the sub-algebra $\mathcal{F} = \mathcal{F}_{\text{QCQP}}^{m,n}(\mathbb{R}^s)$ completes the proof.

## C  PROOF OF PROPOSITIONS IN SECTION 3.3

The two instances are QCQP instances. Both graphs $G$ and $\bar{G}$ consist of the following:

- 6 variable nodes, i.e., $u_j$ or $\bar{u}_j$, where $j \in [6]$. All nodes carry the feature $h^{u_j} = (0, -1, 1)$. here we assume that $-1 \leq x_i \leq 1$ by the unit ball constraint.
- 12 effective quadratic nodes. The squared nodes carry $h^{v_{j,j}} = (0)$, while others carry the feature $h^{v_{j,k}} = (1)$.
- 1 constraint node $c$ representing the unit ball constraint. The node carries feature $(-1)$ for both graphs.

We now verify that the Tripartite WL-test does not separate the two graphs:

- After initialization, we have $h_1^0 := h^{0,u} = h^{0,\bar{u}} = \text{HASH}_1((0, -1, 1))$, $h_2^0 := h^{0,v_{j,j}} = h^{0,\bar{v}} = \text{HASH}_2((0))$, $h_3^0 := h^{0,v_{j,k}} = \text{HASH}_2((1))$ and $h_4^0 := h^{0,c} = h^{0,\bar{c}} = \text{HASH}_3((-1))$.
- After the first sub-iteration, we have
$$\bar{h}_2^0 := \bar{h}^{0,v_{j,k}} = \text{HASH}(h_2^0, 2h_1^0),$$
and
$$\bar{h}_3^0 := \bar{h}^{0,v_{j,k}} = \text{HASH}(h_3^0, 2h_1^0),$$
which remains equal for all $v \in V_2(G)$ and $\bar{v} \in V_2(\bar{G})$.
- After the second sub-iteration, we have
$$h_4^1 := h^{1,c} = h^{1,\bar{c}} = \text{HASH}(h_4^0, 0, 1 \cdot \bar{h}_2^0),$$
which remains equal for both graphs.
- After the third sub-iteration, we have
$$h_2^1 := h^{1,v_{j,j}} = \text{HASH}(\bar{h}_2^0, 1 \cdot h_3^1),$$
and
$$h_3^1 := h^{1,v_{j,k}} = \text{HASH}(\bar{h}_3^0, 0),$$
which remains equal for both graphs.

- After the final sub-iteration, we have

$$h_1^1 := h^{1,u} = h^{1,\bar{u}} = \text{HASH}(h_1^0, 0, 2 \cdot h_2^1 + 1 \cdot h_3^1),$$

  which remains equal for both graphs.
- The Tripartite WL-test terminates after one iteration since no further node pairs are separated.

The Tripartite WL-test returns $C^{0,\cdot}$, which is the same for both instances. Thus, we conclude that the two graphs are not separated, with variables and constraints correspondingly indexed. By Proposition 4, we conclude that, in both the node-level and graph-level cases, tripartite message-passing GNNs cannot separate the two instances.

Therefore, we conclude that tripartite message-passing GNNs cannot approximate the optimal solution or optimal value for non-convex QCQP instances (even QP instances). To demonstrate that tripartite message-passing GNNs cannot accurately predict feasibility, we slightly modify the two instances:

*Proof of Proposition 1.* We reconstruct the objective as a constraint. Specifically, consider the following two instances:

$$
\begin{aligned}
\min \quad & 0 \\
\text{s.t.} \quad & x_1 x_2 + x_2 x_3 + x_3 x_1 + x_4 x_5 + x_5 x_6 + x_6 x_4 \leq -\frac{3}{4} \\
& \sum_i x_i^2 \leq 1
\end{aligned}
\tag{C.1}
$$

and

$$
\begin{aligned}
\min \quad & 0 \\
\text{s.t.} \quad & x_1 x_2 + x_2 x_3 + x_3 x_4 + x_4 x_5 + x_5 x_6 + x_6 x_1 \leq -\frac{3}{4} \\
& \sum_i x_i^2 \leq 1
\end{aligned}
\tag{C.2}
$$

Clearly, instance C.1 is not feasible, while instance C.2 is feasible.

In the graph generated by the Tripartite graph representation, we change the objective to another special constraint and add a new dummy objective. Similarly, we see that tripartite message-passing GNNs fail to separate $\mathcal{I}$ and $\bar{\mathcal{I}}$. $\qquad\square$

