# OpenReview forum: "On Representing Convex Quadratically Constrained Quadratic Programs via Graph Neural Networks"
_ICLR.cc/2025/Conference — Submitted to ICLR 2025_

### Official Review · Reviewer_3k9j · 2024-11-04

**Soundness:** 1
**Presentation:** 2
**Contribution:** 1
**Rating:** 3
**Confidence:** 4

**Summary:**

This paper explores the application of GNNs to QCQP. Specifically, the authors introduce a tripartite graph representation to encode QCQP, apply GNNs to this structure, and analyze their capacity to represent QCQPs. They demonstrate that GNNs can universally represent convex QCQPs but are unable to represent nonconvex QCQPs. Finally, they verify the conclusion with numerical experiments.

**Strengths:**

Extending the application of GNN to QCQP is an interesting direction, and considering nonconvex cases is good -- not just limited to convex cases.

**Weaknesses:**

1. The two counterexamples given in this paper involves only linear constraints, which makes the conclusion less interesting. I understand that linear-constrained QP is a special case of QCQP and "GNN fails on LCQP" implies "GNN fails on QCQP", but I think a "real" QCQP would be the first interest of the potential readers of this paper, as "quadratic constraint" is the major claim in the title and abstract.

2. The paper feels somewhat incomplete. While it highlights the limitations of GNNs on nonconvex QCQPs, it does not propose any potential solutions. The current takeaway seems to be “GNNs are not suitable for nonconvex QCQPs,” which might not be the intended message. Including suggestions or alternative approaches could improve the paper. As nonconvex problems are often studied on a case-by-case basis, identifying a specific nonconvex QCQP scenario where GNNs might still be effective would strengthen the contribution.

3. The numerical experiments are not strong enough. The datasets are generated by perturbing a single instance, offering minimal insights into expressive power and generalization. If the GNN can only handle a single instance (with minor perturbations), it may not truly validate expressive power. Similarly, if the GNN only generalizes to problems perturbed from the sole instance in the training set, it weakly supports generalization.

**Questions:**

Refer to "weaknesses"

---

### Official Review · Reviewer_NtKu · 2024-11-04

**Soundness:** 2
**Presentation:** 2
**Contribution:** 2
**Rating:** 3
**Confidence:** 4

**Summary:**

This paper proposes to represent quadratically constrained quadratic programs (QCQPs) with tripartite graphs. The authors prove that graph neural networks (GNNs) on the tripartite graphs can predict the properties of convex QCQPs. Small-scale numerical results are conducted.

**Strengths:**

1. The authors represent QCQPs with tripartite graphs and prove that GNNs can predict the properties of convex QCQPs.

2. Counter examples are given to show that convexity is necessary.

**Weaknesses:**

1. The theoretical results are not surprising given existing works Chen et al. (2023a), Chen et al. (2023b), and Chen et al. (2024). In fact, the flow of the paper and the proof techniques are very similar to Chen et al. (2023a), Chen et al. (2023b), and Chen et al. (2024).

2. The numerical experiments are limited -- the instances have small sizes and the datasets are not general (they are perturbed from a few instances).

__References:__

(Chen et al., 2023a) Ziang Chen, Jialin Liu, Xinshang Wang, Jianfeng Lu, and Wotao Yin, On representing linear programs by graph neural networks, ICLR 2023.

(Chen et al., 2023b) Ziang Chen, Jialin Liu, Xinshang Wang, Jianfeng Lu, and Wotao Yin, On representing mixed-integer linear programs by graph neural networks, ICLR 2023.

(Chen et al., 2024) Ziang Chen, Xiaohan Chen, Jialin Liu, Xinshang Wang, and Wotao Yin, Expressive Power of Graph Neural Networks for (Mixed-Integer) Quadratic Programs, arXiv: 2406.05938.

**Questions:**

None

---

### Official Review · Reviewer_MG1L · 2024-11-04

**Soundness:** 4
**Presentation:** 4
**Contribution:** 3
**Rating:** 6
**Confidence:** 4

**Summary:**

The paper introduces a new tripartite graph representation for QCQPs, demonstrating that this structure has strong representational power. It shows that, for a given space of QCQP models, there exists such a network that can accurately distinguish all the models from the space. Additionally, the authors present numerical experiments with message-passing GNNs to validate the effectiveness of the approach.

**Strengths:**

This paper introduces a new graph neural network architecture that improves upon existing models for optimization problems, specifically targeting quadratically constrained quadratic programs. The analysis of the universal approximation properties is solid. The paper is well-written and addresses the QCQP problem that has been largely unexplored in the literature.

**Weaknesses:**

a. The new tri-partite network contains O(n^2) nodes in each layer, where n is the number of variables. This represents a significant increase in computational cost compared to traditional GNNs, which have O(n) nodes per layer.

b. While QCQP problems have applications across various industries, the experiments in this paper appear limited and address only small-scale examples. The authors could comment on this gap.

**Questions:**

Since the tri-partite graph has O(n^2) nodes in each layer, it may be comparable to existing but more complex networks such as second order folklore GNNs. In terms of the network size and representational power, what are the advantages of the proposed tri-partite graph compared to second order folklore GNNs?

---

### Official Review · Reviewer_o7kh · 2024-11-04

**Soundness:** 3
**Presentation:** 2
**Contribution:** 3
**Rating:** 5
**Confidence:** 3

**Summary:**

This paper introduces a tri-partite graph to represent convex quadratically constrained quadratic programming (QCQP) instances and a corresponding message passing graph neural network (MP-GNN) to approximate property mappings of the programming problem. The presentation style resembles Chen et al'24 for QCLP. The paper is a nice addition to the literature.

**Strengths:**

The tri-partite graph representation is new. The claim of Theorem 1 is impressive. Nonconvex counter examples are presented.

**Weaknesses:**

The results of the paper are presented without in-depth comparison and discussions that are necessary to argue the chosen graph representations are simplest possible.

The numerical examples are limited to training performance and up to only mid-sized feasible QCQPs, which classic solvers are also capable of solving.

**Questions:**

1. When a QCQP comes without quadratic constraints, it reduces to a QCLP. Is there any advantages and disadvantages of the approach in this paper compared to Chen et al. 2024?

2. What has limited the practical performance of the proposed method?

---

### Meta-Review · Area_Chair_GZig · 2024-12-20

**Metareview:**

This paper uses a tri-partite graph to represent QCQP instances, and it implements a message-passing graph neural network on this graph. It provides numerical experiments to illustrate the performance of the method.
The reviewers appreciate the tri-partite representation of the problem, but they raised concerns about the computational complexity and the small scale of the numerical experiments.

**Additional Comments On Reviewer Discussion:**

The authors provided further numerical experiments during the rebuttal period. Some of the reviewers acknowledged their response but they did not change their scores.

---

### Decision · Program_Chairs · 2025-01-22

Reject